# Tiny-R1V: Lightweight Multimodal Unified Reasoning Model via Model Merging

## Abstract

Although Multimodal Large Language Models (MLLMs) have demonstrated remarkable capabilities across diverse tasks, they encounter numerous challenges in terms of reasoning efficiency, such as large model size, overthinking, and compromised accuracy in lightweight scenarios. However, research on the reasoning capabilities of lightweight MLLMs is quite lacking. To this end, we propose Tiny-R1V, a novel lightweight 3B model that achieves faster inference and higher accuracy via a two-stage optimization, while unifying multimodal reasoning across multiple tasks and using fewer tokens. In the first stage, Tiny-R1V introduces Length-Informed Relative Policy Optimization (LIPO), a novel reinforcement learning method, to train each reasoning model. The LIPO is designed to dynamically adjusts advantages of responses within groups, that is, by prioritizing concise yet high-quality responses to encourage the generation of shorter and more accurate response. In the second stage, we propose Adaptive Model Merging (AMM), a training-free model merging method that merges multiple specialist models into a unified architecture. Specifically, AMM adaptively adjusts the weights of task vectors and robustly optimizes the merged vectors via a novel gradient projection regularization loss function, thus mitigating redundant conflicts between them. Extensive evaluations on ten widely-used reasoning benchmarks covering mathematics, structured data (charts, tables, documents), OCR, and general capabilities showcase the superior performance of Tiny-R1V, enabling lightweight models to excel in diverse multimodal reasoning tasks.

## 1 Introduction

Multimodal Large Language Models (MLLM) (Bai et al., 2025; Hurst et al., 2024; Team et al., 2023; Yao et al., 2024b) have shown powerful capabilities in the extensive application across different tasks. However, MLLMs still encounter several challenges in terms of reasoning ability. On the one hand, the improvement of the models' reasoning ability is constrained by scaling laws (Kaplan et al., 2020), making it impossible to significantly boost model reasoning performance with smaller parameters. On the other hand, they also confront issues such as large-scale training costs and the balance in the joint training of different reasoning tasks.

The recent success of Reinforcement Learning (RL) in Large Language Models(LLMs), such as Kimi-K1.5 (Team et al., 2025) and DeepSeek-R1 (Guo et al., 2025) have demonstrated its potential in motivating the models' long chain-of-thought(CoT) reasoning ability through rule-based Group Relative Policy Optimization (GRPO) (Shao et al., 2024), enabling LLMs to handle complex reasoning tasks. However, due to long CoT, the existing reasoning models will inevitably incur high inference costs and suffer from the issue of over-thinking (Sui et al., 2025; Chen et al., 2024b), which restricts their deployment in real-time or resource-constrained scenarios. Therefore, to reduce the redundancy of long-CoT, some efficiency-oriented methods are proposed through pruning (Luo et al., 2025) or compression (Chen et al., 2024b), while they fundamentally overlook the potential for greater gains from shorter reasoning CoT.

Moreover, in the field of traditional vision tasks and LLMs (Akiba et al., 2025; Ilharco et al., 2022; Cheng et al., 2025), model merging techniques have emerged as a powerful training-free approach to combine multiple specialist models into a unified architecture, which retains the capabilities of each expert model while improving the unified model's understanding ability. Hence, corresponding

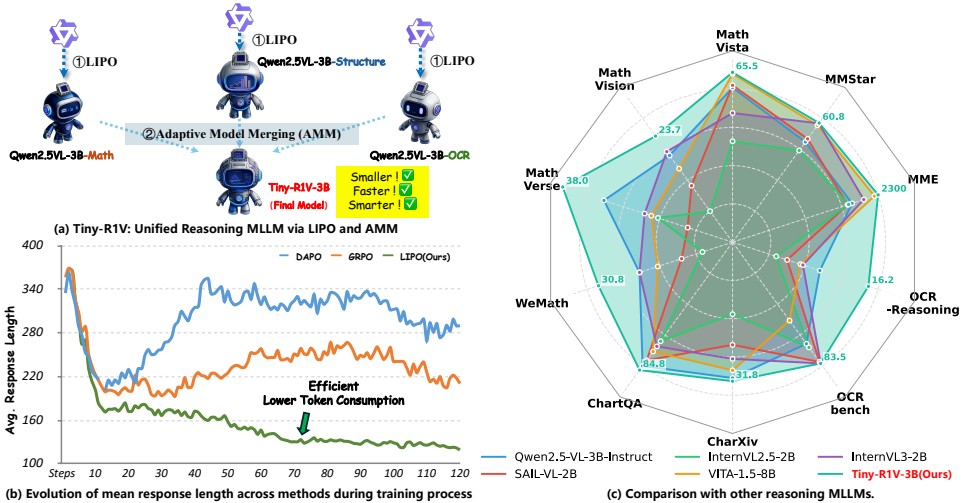

Figure 1: (**a**) Two-stage framework for training lightweight MLLMs for unified reasoning tasks. (**b**) The average response length of GRPO (Shao et al., 2024), DAPO (Yu et al., 2025) and LIPO(Ours) on the training set during the RL training process. (**c**) Tiny-R1V achieves the state-of-the-art performance on a broad range of multimodal reasoning tasks compared with other open source models.

to the above progress, a critical question emerges: ***Is it possible to devise a novel paradigm that empowers lightweight models to achieve efficient and accurate reasoning across diverse tasks?***

In this paper, as shown in Figure 1(a), we propose a lightweight 3B model named Tiny-R1V, which achieves both faster inference speed and more accurate reasoning performance via joint a two-stage learning framework. In the first stage, for reinforcement learning, we provide large-scale training data encompassing various tasks such as geometry, chart, table and OCR. For each task, we collect public datasets containing at least 10k samples to ensure effective post-training via reinforcement learning. Building upon this, we design a novel Length-Informed Relative Policy Optimization (LIPO) to dynamically adjust the advantages of responses within the group, which aims to reduce the advantages of longer responses and increase those of shorter ones among responses that have approximately equivalent rewards. In this manner, as shown in Figure 1(b), each response is constrained within a valid range, while ensuring that the model can output more accurate responses in the form of concise answers as much as possible.

In the second stage, we propose a novel model merging method, namely Adaptive Model Merging (AMM), aiming to enhance the optimization of task vectors (i.e., parameter changes between post-training models and the base model). Specifically, AMM adaptively adjusts the weights of task vectors through inherent task importance parameters $\alpha$ and dynamic state parameters $\beta$, and robustly optimizes the merged vectors via the gradient projection regularization loss function. Meanwhile, AMM enables the integration of multiple MLLMs without requiring additional data for training. Furthermore, as shown in Figure 1(c), the proposed merging method effectively consolidates inputs from diverse tasks and outperforms state-of-the-art (SOTA) models trained on mixed training data. In summary, the main contributions of this work are summarized as follows:

1. We propose a two-stage framework for training lightweight multimodal models for reasoning tasks, and then construct Tiny-R1V with only 3B parameters.

2. We introduce Length-Informed Relative Policy Optimization (LIPO), which dynamically adjusts the inter-group response advantages, minimizing the number of response tokens while ensuring the accuracy of the answer.

3. We design a novel Adaptive Model Merging (AMM) , which not only retains the unique advantages of each model, but also reduces the redundant interference between models during the model merge process.

4. Extensive experiments on ten MLLM reasoning benchmarks derived from four different tasks demonstrate the superiority of our proposed Tiny-R1V.

## 2 RELATED WORK

### 2.1 REINFORCEMENT LEARNING FOR MLLMS REASONING CAPABILITY

Reinforcement learning(RL) (Cao et al., 2024) has been proven to be a key technology for enhancing the reasoning capabilities of LLMs. Early research mainly focused on Reinforcement Learning from Human Feedback (RLHF) (Bai et al., 2022; Yu et al., 2024b). However, RLHF's reliance on high-quality manual annotations limits its scalability in larger-scale scenarios. DeepSeek-R1 (Guo et al., 2025) propose a minimalist rule-based reward function, which can provide reliable reward for reinforcement learning without complex human annotations. Inspired by this, recent studies have begun to introduce reinforcement learning into the multimodal (Yao et al., 2025a; 2024a; Zhang et al., 2025a; Yao et al., 2025b) domain to boost model performance in complex visual reasoning tasks. Existing works (Yang et al., 2025; Liu et al., 2025) have designed fine-grained rule-based reward functions from various perspectives, such as answer correctness (Peng et al., 2025), reasoning chain completeness (Zhang et al., 2025a), and visual consistency (Huang et al., 2025b), which effectively improves the accuracy and robustness of MLLMs reasoning capability.

### 2.2 MODEL MERGING

Model merging (Yang et al., 2024) emerges as a cost-effective and flexible strategy to enable the integration of capabilities from multiple expert models without additional training. The training-free weight merging method assumes that all expert models share the same initialization parameters, and directly applies strategies such as linear interpolation (Wortsman et al., 2022; Ilharco et al., 2022), sparsification (Yadav et al., 2023; Yu et al., 2024a), and low-rank optimization to the parameters in the weight space (Choi et al., 2024; Cheng et al., 2025; Wei et al., 2025; Miyano & Arase, 2025; Zhang et al., 2025b) to achieve parameter merging. Meanwhile, inspired by the idea of the Mixture-of-Experts (MoE) architecture, the dynamic routing merging (Tang et al., 2024; Muqeeth et al., 2023) method dynamically calculates the weights of each expert model based on input samples during the inference phase and generates combination coefficients in real-time through a lightweight routing mechanism.

## 3 METHOD

This section first provides the preliminaries, then introduces the key techniques of the proposed Tiny-R1V framework, namely **L**ength-**I**nformed Relative **P**olicy **O**ptimization (**LIPO**) and **A**daptive **M**odel **M**erging (**AMM**).

### 3.1 PRELIMINARIES

The GRPO (Shao et al., 2024)framework initially leverages a MLLM to initialize both a policy model $\pi_\theta$ and a reference model $\pi_{\text{old}}$. For a given image-text pair $(\mathcal{I}, \mathcal{T})$, the reference policy model $\pi_{\theta_{\text{old}}}$ generates $G$ responses $\{o_1, o_2, ..., o_G\}$. A group-based reward function then computes the corresponding rewards $\{R_1, R_2, ..., R_G\}$, which are subsequently utilized to estimate the advantage $\hat{A}i$ for each response within the group, $\hat{A}_i = (R_i - \text{mean}(\{R_i\}_{i=1}^G))/\text{std}(\{R_i\}_{i=1}^G)$. Then, GRPO employs a clipped objective with a KL penalty term:

$$\mathcal{J}_{\text{GRPO}}(\theta) = \mathbb{E}_{\substack{(\mathcal{I},\mathcal{T})\sim\mathcal{D}, \\ o\sim\pi_{\theta_{\text{old}}}(\cdot|\mathcal{I},\mathcal{T})}}$$

$$\left[\frac{1}{n}\sum_{i=1}^n \min\left(\frac{\pi_\theta(o_i \mid \mathcal{I},\mathcal{T})}{\pi_{\theta_{\text{old}}}(o_i \mid \mathcal{I},\mathcal{T})}\hat{A}_i, \text{clip}\left(\frac{\pi_\theta(o_i \mid \mathcal{I},\mathcal{T})}{\pi_{\theta_{\text{old}}}(o_i \mid \mathcal{I},\mathcal{T})}, 1-\epsilon, 1+\epsilon\right)\hat{A}_i - \beta D_{\text{KL}}\left(\pi_\theta \mid\mid \pi_{\text{ref}}\right)\right)\right] \tag{1}$$

**WUDI Model Merging.** Given a pre-trained base model $\theta_0$ and $P$ task-specific models $\{\theta_i\}_{i=1}^P$, the task vector for each task $i$ is defined as the parameter difference between the task-specific model and the base model, $\tau_i = \theta_i - \theta_0$. WUDI-merging (Cheng et al., 2025) aims to minimize the interference between the merged task vector and each individual task vector, as $\tau_{m,l} - \tau_{i,l}$ for task $i$ at layer $l$. Intuitively, it encourages the merged task vector to retain the key information of each task vector

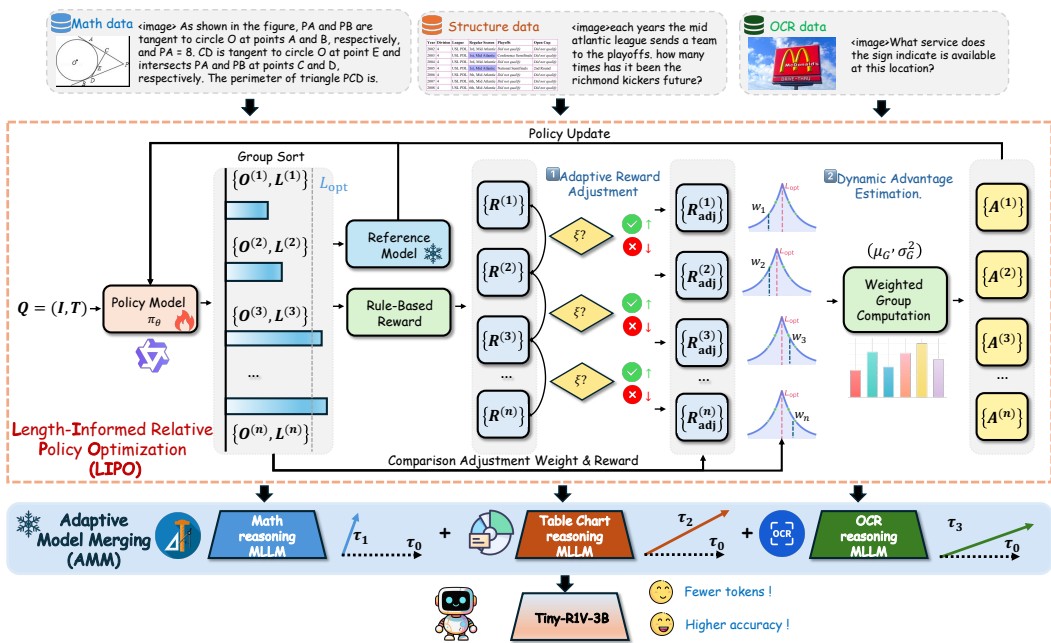

Figure 2: **Overview of the proposed Tiny-R1V**. Tiny-R1V employs Length-Informed Relative Policy Optimization (LIPO) and Adaptive Model Merging (AMM). In the first stage, Tiny-R1V trains three expert models separately using LIPO, which dynamically adjusts the advantages between groups to minimize the number of response tokens while ensuring the accuracy of the answer. In the second stage, Tiny-R1V merges the three models using AMM, determines the dynamic weights of model parameters, and reduces parameter conflicts, resulting in the final Tiny-R1V-3B model.

in its own direction, thereby reducing redundancy and conflict between tasks. For each layer $l$, the merged task vector is iteratively refined over $\mathcal{N}$ steps using a specialized loss function:

$$\min_{\boldsymbol{\tau}_{m,l}} \mathcal{L}_l = \sum_{i=1}^{n} \frac{1}{\|\boldsymbol{\tau}_{i,l}\|_F^2} \left\| (\boldsymbol{\tau}_{m,l} - \boldsymbol{\tau}_{i,l}) (\boldsymbol{\tau}_{i,l})^\top \right\|_F^2 \tag{2}$$

where $\| \cdot \|_F$ denotes the Frobenius (Böttcher & Wenzel, 2008) norm of a matrix.

## 3.2 LIPO: LENGTH-INFORMED RELATIVE POLICY OPTIMIZATION

To achieve lightweight model reasoning, relieve the problem of overthinking, and favor simple yet correct responses, we propose Length-Informed Relative Policy Optimization (**LIPO**), a novel online MLLM reinforcement learning framework. It rewards shorter and correct response through adaptive reward adjustment and dynamic advantage estimation.

For a given question $q \in \mathcal{Q}$ and $k$ responses generated by the model $\mathcal{O} = \{o^{(1)}, o^{(2)}, \ldots, o^{(k)}\}$, the length of each response is $L = \{L^{(1)}, L^{(2)}, \ldots, L^{(k)}\}$, a response $o^* \in \mathcal{O}$ is Pareto optimal (Ngatchou et al., 2005) if and only if there does not exists any other response $o \in \mathcal{O}$ such that at least one of the following conditions holds $Q(o) \geq Q(o^*)$ or $L(o) \leq L(o^*)$, within the feasible solution set in the quality-length (Q-L) space $\mathcal{F} = \{(Q(o), L(o)) \mid o \in \mathcal{O}\}$.

**Adaptive Reward Adjustment.** The core motivation is to correct the $Q$-$L$ imbalance problem of the traditional reward mechanism, explicitly favoring concise yet high-quality responses while ensuring meaningful reasoning length. Responses within a group are sorted by length in ascending order as $L^{(1)} \leq L^{(2)} \leq \cdots \leq L^{(k)}$. For a pair of adjacent responses to qualify for reward adjustment, $(L^{(i)}, R^{(i)})$ and $(L^{(i+1)}, R^{(i+1)})$, they must satisfy a trigger condition $\xi(o^{(i)}, o^{(i+1)})$, defined as $\xi(o^{(i)}, o^{(i+1)}) = \left(|R^{(j)} - R^{(i)}| < \eta\right) \wedge \left(L^{(i+1)} - L^{(i)} > 0\right) \wedge \left(L^{(i)} \geq L_{\min}\right)$. This condition

encodes three critical constraints:$|R^{(j)} - R^{(i)}| < \eta$ ensures that the difference in reward values between two adjacent responses is within a certain threshold $\eta$, indicating that their qualities are relatively close. $L^{(i+1)} - L^{(i)} > 0$ and $L^{(i)} \geq L_{\min}$ ensure that response with appropriate length are selected through comparison. Based on this trigger condition, we adjust the reward for the shorter response $R^{(i)}$ as follows:

$$R_{\text{adj}}^{(i)} = \begin{cases} R^{(i)}(1+\alpha) & \xi \wedge L^{(i)} < L_T \\ R^{(i)}(1+\alpha\omega)) & \xi \wedge L^{(i)} \geq L_T \\ R^{(i)} & \text{otherwise} \end{cases} \tag{3}$$

where, $\alpha \in (0, 0.2]$ is a base enhancement factor, and $\omega$ is a decay term that modulates the enhancement, defined as $\omega(o^{(i)}, o^{(i+1)}) = \max\left(0, 1 - \frac{L^{(i)} - L_T}{L^{(i+1)} - L_T + \epsilon}\right)$. For responses significantly shorter than a threshold $L_T$, we apply the full enhancement of $\alpha$ to strongly incentivize such concise outputs. For responses whose lengths are longer than $L_T$, when $L^{(i)}$ approaches $L^{(i+1)}$, it means that the shorter response is only slightly shorter than the longer one. At this time, $\omega$ tends 0, thus reducing the enhancement. In summary, this design achieves a smooth transition between full and reduced reinforcement, and balance the reward signal and the simplicity of the response.

**Dynamic Advantage Estimation.** We propose a dynamic advantage estimation to normalize the reward by calculating the advantage of each response relative to the ideal length within the group. The ideal length of each response group is defined as $L_{\text{opt}} = \max(2L_{\min}, \text{median}(L^{(i)}))$. By taking the maximum of these two values, $L_{\text{opt}}$ ensures that the optimal length is both practical and representative of the group's characteristics. Then, the weight $w_i$ for each response is computed as:

$$w_i = \exp\left(-\phi|L^{(i)} - L_{\text{opt}}|\right) \tag{4}$$

where $\phi$ is the scaling factor and the weight $w_i$ reflects the proximity of each response's length to the optimal length $L_{\text{opt}}$. Responses that are closer to $L_{\text{opt}}$ will be assigned higher weights, emphasizing their importance in the advantage estimation process. The dynamic advantage $A^{(i)}$ is then calculated using the following set of equations:

$$A^{(i)} = \frac{R_{\text{adj}}^{(i)} - \mu_G}{\sigma_G + \epsilon} \tag{5}$$

where, $\mu_G = \frac{\sum_{i \in G} w_i R_{\text{adj}}^{(i)}}{\sum_{i \in G} w_i}$ is the weighted average of the adjusted rewards, $\sigma_G^2 = \frac{\sum_{i \in G} w_i (R_{\text{adj}}^{(i)} - \mu_G)^2}{\sum_{i \in G} w_i}$ is the weighted variance. This dynamic advantage estimation helps the model better understand the relative quality of each response within the group, guiding it to generate more optimal answers.

**Reward function.** Consistent with GRPO, LIPO also adopts rule-based rewards, with exact matching rewards for maths tasks and structured data (table, chart, document) tasks. For OCR tasks, a more continuous Levenshtein Distance (Yujian & Bo, 2007) is utilized, where the reward is calculated by comparing the edit distance between the response string and the target string.

## 3.3 **AMM**: ADAPTIVE MODEL MERGING

During model merging, different tasks have different importance and compatibility. WUDI Merging (Cheng et al., 2025) uses fixed weights, which makes it difficult to effectively balance multi-task performance. We propose a dual-weight adaptive mechanism to achieve refined control of task vectors by introducing two weight parameters, $\alpha$ and $\beta$. The $\alpha_{i,l} = \frac{\|\boldsymbol{\tau}_{i,l}\|_F^2}{\sum_{j=1}^n \|\boldsymbol{\tau}_{j,l}\|_F^2}$ parameter measures the inherent importance of tasks based on the norm of the task vector to ensure that key tasks are not neglected. The $\beta_{i,l}^n$ parameter is dynamically adjusted based on the current merge state. It is used to measure the compatibility of tasks with the merge vector and focus on the tasks that match the current merge result best.

$$\beta_{i,l}^n = \exp\left(-\gamma \cdot \frac{\|(\boldsymbol{\tau}_{m,l}^{n-1} - \boldsymbol{\tau}_{i,l})\boldsymbol{\tau}_{i,l}^\top\|_F^2}{\|\boldsymbol{\tau}_{i,l}\|_F^2}\right) \tag{6}$$

where $\gamma$ is the scaling factor. The calculation of $\beta_{i,l}^n$ is based on the merged vector $\boldsymbol{\tau}_{m,l}^{n-1}$ after the $(n-1)$th iteration, so that it will be dynamically adjusted according to the latest state of the merged

vector. As the iteration proceeds, the merged vector $\boldsymbol{\tau}_{m,l}$ gradually converges, and $\beta_{i,l}^n$ will also tend to be stable, thus the merged vector can retain the key information of multiple tasks at the same time.

The original merge loss $\mathcal{L}_l$ in Eq. 2 only focuses on the compatibility of the merged vector with each task vector. However different directions in the parameter space have different impacts on task performance, and a slight change in some parameter directions may significantly affect task accuracy. Therefore, we design the gradient projection regularization $\mathcal{R}_l^n$, which decomposes the gradient $\nabla\mathcal{L}_l^n$ into two components. The parallel component $\frac{\nabla\mathcal{L}_l^n \cdot \boldsymbol{\tau}_{i,l}^\top}{\|\boldsymbol{\tau}_{i,l}\|_F^2}\boldsymbol{\tau}_{i,l}$, which represents the update consistent with the direction of the task vector $\boldsymbol{\tau}_{i,l}$. And the orthogonal component $\nabla\mathcal{L}_l^n - \frac{\nabla\mathcal{L}_l^n \cdot \boldsymbol{\tau}_{i,l}^\top}{\|\boldsymbol{\tau}_{i,l}\|_F^2}\boldsymbol{\tau}_{i,l}$, which represents the update perpendicular to the direction of the task vector (causing task interference). By penalizing the norm of the orthogonal component, the optimization process is forced to proceed along the direction of the task vector, thus reducing the updates that are harmful to specific tasks. The penalty term $\mathcal{R}_l^n$ is defined as follows:

$$\mathcal{R}_l^n = \lambda \cdot \sum_{i=1}^{\mathcal{P}} \left\| \nabla\mathcal{L}_l^n - \frac{\nabla\mathcal{L}_l^n \cdot \boldsymbol{\tau}_{i,l}^\top}{\|\boldsymbol{\tau}_{i,l}\|_F^2}\boldsymbol{\tau}_{i,l} \right\|_F^2 \tag{7}$$

The algorithm flow is described in detail in Algorithm 1. The optimization of each linear layer is independent, and the problem can be solved layer by layer in turn by using the gradient descent method for optimization.

---

**Algorithm 1** AMM: Adaptive Model Merging

---

1: **Input:** Parameters $\boldsymbol{\theta}$; task vectors $\mathcal{T} = \{\boldsymbol{\tau}_{i,l}\}_{i=1}^{\mathcal{P}}$; steps $\mathcal{N}$; learning rate $\zeta$; regularization strength $\gamma$; gradient regularization strength $\lambda$
2: **Output:** Merged parameters $\boldsymbol{\theta_m}$.
3: $\triangleright$ *Initialize with weighted sum* $(\star)$
4: **for** linear layer $l \in \{1, \cdots, \Psi\}$ **do**
5:     Compute $\alpha_{i,l}, \beta_{i,l}^0$; $\boldsymbol{w}_{i,l}^0 = \alpha_{i,l} \cdot \beta_{i,l}^0 / \sum_{j=1}^{\mathcal{P}}(\alpha_{j,l} \cdot \beta_{j,l}^0)$; $\boldsymbol{\tau}_{m,l}^0 = \sum_{i=1}^{\mathcal{P}} \boldsymbol{w}_{i,l}^0 \cdot \boldsymbol{\tau}_{i,l}$
6: **end for**
7: **for** linear layer $l \in \{1, \cdots, \Psi\}$ **do**
8:     **for** $n \in \{1, \cdots, \mathcal{N}\}$ **do**
9:         $\triangleright$ *Update dynamic weights* $(\star)$
10:         $\beta_{i,l}^n = \exp\left(-\gamma \cdot \frac{\|(\boldsymbol{\tau}_{m,l}^{n-1} - \boldsymbol{\tau}_{i,l})\boldsymbol{\tau}_{i,l}^\top\|_F^2}{\|\boldsymbol{\tau}_{i,l}\|_F^2}\right)$; $\boldsymbol{w}_{i,l}^n = \alpha_{i,l} \cdot \beta_{i,l}^n / \sum_{j=1}^{\mathcal{P}}(\alpha_{j,l} \cdot \beta_{j,l}^n)$
11:         $\triangleright$ *Weighted loss + projection regularization* $(\star)$
12:         $\mathcal{L}_l^n = \sum_{i=1}^{\mathcal{P}} \frac{\boldsymbol{w}_{i,l}^n}{\|\boldsymbol{\tau}_{i,l}\|_F^2} \left\|(\boldsymbol{\tau}_{m,l}^{n-1} - \boldsymbol{\tau}_{i,l})\boldsymbol{\tau}_{i,l}^\top\right\|_F^2$; $\mathcal{R}_l^n = \lambda \cdot \sum_{i=1}^{\mathcal{P}} \left\|\nabla\mathcal{L}_l^n - \frac{\nabla\mathcal{L}_l^n \cdot \boldsymbol{\tau}_{i,l}^\top}{\|\boldsymbol{\tau}_{i,l}\|_F^2}\boldsymbol{\tau}_{i,l}\right\|_F^2$
13:         $\triangleright$ *Regularized update* $(\star)$
14:         $\boldsymbol{\tau}_{m,l}^n = \boldsymbol{\tau}_{m,l}^{n-1} - \zeta \cdot \left(\nabla\mathcal{L}_l^n + \nabla_{\boldsymbol{\tau}_{m,l}^{n-1}}\mathcal{R}_l^n\right)$
15:     **end for**
16: **end for**
17: $\triangleright$ Assemble merged task vectors
18: $\boldsymbol{\tau}_m = \{\boldsymbol{\tau}_{m,l}\}_{l=1}^{\Psi}$, $\boldsymbol{\theta_m} = \boldsymbol{\theta} + \boldsymbol{\tau}_m$

---

## 4 EXPERIMENTS

In this section, we first provide the experiments setup in Sec. 4.1, and then present main results in Sec 4.2 that demonstrate the effectiveness of Tiny-R1V. In Sec. 4.3, we conduct ablation studies to evaluate the impact of each design in Tiny-R1V. Sec. 4.4 provides qualitative results of Tiny-R1V.

### 4.1 EXPERIMENTS SETUP

**Implementation.** In this work, we adopt Qwen2.5-VL-3B as our base model. Model training is implemented using the EasyR1 (Yaowei Zheng, 2025) codebase, and the training is executed on 8

Table 1: Composition of our aggregated dataset from public sources, with their corresponding sizes.

| Task | Size | Datasets |
|------|------|----------|
| **Math** | 15K | Geometry3K (Lu et al., 2021) GeoQA+ (Cao & Xiao, 2022) K12 (Meng et al., 2025) UniGeo (Chen et al., 2022) |
| **Table Chart Document** | 15K | TAT-DQA (Zhu et al., 2022) WTQ (Pasupat & Liang, 2015) TabFact (Chen et al., 2019) PlotQA (Methani et al., 2020) TQA (Kembhavi et al., 2017) ChartGalaxy (Li et al., 2025) |
| **OCR** | 10K | TextVQA (Singh et al., 2019) ChromeWriting TextOCR (Singh et al., 2021) OCR-VQA (Mishra et al., 2019) |
| **All(Mix)** | 35K | - |

NVIDIA A800 (40G) GPUs. For the rollout parameters, we set the number of samples per question to 5 and a probability $p$ of 0.3. Regarding RL-related hyperparameters, we use a global batch size of 128, a rollout batch size of 512, a rollout temperature of 0.7, and a learning rate of 1e-6. For training data, we collect a broader range of domain-specific data, which is divided into math, structured data (table, chart, document) and OCR. The datasets used are summarized in Table 1.

**Evaluation Benchmark.** Our model is evaluated across three key dimensions, multimodal mathematical reasoning, multimodal structured data reasoning, and OCR capabilities. For multimodal mathematical reasoning, we compare detailed performance metrics on the MathVista (MINI) (Lu et al., 2023), MathVision (Wang et al., 2024a), MathVerse (Vision-Only) (Zhang et al., 2024), and WeMath (Strict) (Qiao et al., 2024) benchmarks. In terms of multimodal structured data reasoning, evaluations are conducted on ChartQA (Test Average) (Masry et al., 2022) and CharXiv (Reasoning Questions) (Wang et al., 2024b) benchmarks. For OCR capabilities, the model is assessed using the OCRbench (Liu et al., 2024) and OCR-Reasoning (Huang et al., 2025a) benchmarks. Additionally, to verify that the Tiny-R1V model retains general capabilities, we also report its performance on the MME (Fu et al., 2023) and MMStar (Chen et al., 2024a) benchmarks.

## 4.2 MAIN RESULTS

To comprehensively evaluate the effectiveness of our proposed Tiny-R1V, we conduct extensive comparisons across 10 widely used and challenging benchmarks, as illustrated in Table 2.

**Individual models.** Without using cold start, we apply the data in Table 1 and use LIPO to train math model, structure model and OCR model, respectively. LIPO yields a substantial enhancement in the reasoning capabilities of MLLMs. For example, in challenging math benchmarks such as MathVista and MathVerse, Math Model (with LIPO) achieves +4.2% and +4.6% improvement, respectively. In the challenging reasoning chart understanding benchmark CharXiv, structure Model (with LIPO) obtains +3.2% improvement. On the OCR-Reasoning task that examines both perception and reasoning capabilities, the three models (with LIPO) improve by +4.4%, +2.3%, and +1.1%, respectively. It is worth noting that although we do not specifically train a model for general ability, we also achieve improvement on the general ability benchmarks MME and MMStar, which shows the generalization ability of LIPO in enhancing reasoning ability across different tasks.

**Comparison with other merging methods.** We further compare Tiny-R1V with representative model merging methods. The results show that Tiny-R1V outperforms all the compared merging methods across the board, achieving the highest average score of 51.6, which is +0.8% higher than the second-best method namely WUDI Merging, and +4.9% higher than the baseline Task Arithmetic. This constitutes a notable improvement, signifying that the model achieves a distinct average enhancement across 10 benchmarks while fully preserving its various reasoning capabilities. This indicates that our approach effectively merges specialized capabilities, and well alleviates the problem of unbalanced performance across different tasks that exists in other merging methods.

**Comparison with other models.** We compare Tiny-R1V with other general MLLMs. With less training data, our Tiny-R1V outperforms models such as InternVL2.5-2B and VITA-1.5-8B, with an average performance improvement of +6.7%. It is worth noting that in addition to having reasoning ability, Tiny-R1V also demonstrates stronger generalization capabilities in different tasks.

## 4.3 ABLATION STUDY

**Response length discussion.** The following analysis takes mathematical models as an example to illustrate the effectiveness of the LIPO method in reducing inference tokens. Table 3 clearly demon-

Table 2: Capability merging results on Qwen2.5-VL-Instruct (RL post training) across multiple tasks. The best and second best average results are highlighted in boldface and underlined respectively. * denotes evaluation on official open-source weights using VLMEvalKit (Duan et al., 2024).

| Methods | Math | | | | Table & Chart &Doc | | OCR | | General | | Avg. |
|---|---|---|---|---|---|---|---|---|---|---|---|
| | MathVista (mini) | MathVision (full) | MathVerse (O Vision) | WeMath (Strict) | ChartQA (test Avg.) | CharXiv (RQ) | OCRbench | OCR-Reasoning | MME (sum) | MMStar | |
| Qwen2.5-VL-3B-Instruct (Bai et al., 2025) | 62.3 | 21.2 | 31.2 | 22.9 | 84.0 | 31.3 | 79.7 | 12.2 | 2157 | 55.9 | 47.8 |
| *Individual models* | | | | | | | | | | | |
| **Individual Math Model-LIPO (Ours)** | 66.5 | 23.3 | 35.8 | 30.7 | 83.9 | 32.0 | 73.5 | 16.6 | 2304 | 58.5 | 50.3 |
| **Individual Structure Model-LIPO (Ours)** | 61.9 | 22.0 | 33.1 | 23.6 | 85.8 | 34.5 | 77.6 | 14.5 | 2302 | 58.7 | 49.4 |
| **Individual OCR Model-LIPO (Ours)** | 57.9 | 20.0 | 31.9 | 26.3 | 83.5 | 30.4 | 84.6 | 13.3 | 2254 | 57.4 | 48.6 |
| *Comparison with other merging methods* | | | | | | | | | | | |
| Task Arithmetic (Ilharco et al., 2022) | 62.3 | 21.6 | 28.6 | 26.6 | 78.2 | 30.1 | 77.3 | 12.0 | 2044 | 56.6 | 46.7 |
| TA+Dare (Yu et al., 2024a) | 62.2 | 21.8 | 28.5 | 26.6 | 79.6 | 31.1 | 76.2 | 12.3 | 2034 | 56.8 | 46.8 |
| TIES Merging (Yadav et al., 2023) | 61.6 | 22.2 | 33.2 | 28.4 | 83.9 | 31.3 | 78.0 | 13.2 | 2298 | 58.0 | 49.2 |
| TIES+Dare (Yu et al., 2024a) | 62.6 | 23.0 | 28.9 | 28.9 | 84.0 | 31.6 | 78.4 | 13.2 | 2312 | 58.2 | 49.6 |
| TSV Merging (Gargiulo et al., 2025) | 61.5 | 23.5 | 34.9 | 30.6 | 85.0 | 31.4 | 79.6 | 14.0 | 2269 | 59.5 | 50.1 |
| Iso-C Merging (Marczak et al., 2025) | 62.1 | 22.2 | 34.9 | 29.5 | 84.9 | 32.2 | 80.0 | 13.3 | 2286 | 57.6 | 49.8 |
| WUDI Merging (Cheng et al., 2025) | 63.5 | 23.3 | 37.8 | 30.3 | 85.0 | 32.4 | 81.5 | 15.2 | 2262 | 58.7 | 50.8 |
| WUDI Merging v2 (Wei et al., 2025) | 61.8 | 23.4 | 34.7 | 31.4 | 85.0 | 32.2 | 79.8 | 15.0 | 2269 | 59.5 | 50.4 |
| *Comparison with mixture training* | | | | | | | | | | | |
| Mixture Training | 64.5 | 23.2 | 34.2 | 30.0 | 84.8 | 32.0 | 82.0 | 12.8 | 2269 | 59.5 | 50.4 |
| *Comparison with other models* | | | | | | | | | | | |
| SAIL-VL-2B (Dong et al., 2025) | 62.8 | 17.3 | 17.4 | 14.8 | 82.9 | 26.1* | 83.2 | 9.5* | 2132* | 56.7 | 44.7 |
| InternVL2.5-2B (Chen et al., 2024c) | 51.1 | 14.0 | 22.3 | 10.8 | 79.2 | 21.3 | 80.4 | 8.6* | 2138 | 53.7 | 41.8 |
| InternVL3-2B (Zhu et al., 2025) | 57.0 | 21.7 | 24.5 | 22.9 | 80.2 | 28.3 | 83.5 | 10.8 | 2221 | 60.7 | 46.9 |
| VITA-1.5-8B (Fu et al., 2025) | 65.2* | 19.5 | 23.4 | 19.4 | 81.2* | 30.1* | 75.2 | 10.6* | 2280* | 60.2 | 46.6 |
| **Tiny-R1V-3B (Ours)** | **65.5** | **23.7** | **38.0** | **30.8** | **85.3** | **32.4** | **82.5** | **16.2** | **2291** | **59.5** | **51.6** |

strates the remarkable balance between the accuracy and response efficiency achieved by our LIPO-based models. The Qwen2.5-VL-3B model with GRPO consumes average 138.1 tokens to reach 66.1% accuracy on MathVista, whereas our LIPO-enabled Qwen2.5-VL-3B-Instruct and Tiny-R1V-3B models deliver superior or comparable accuracy (66.5% and 65.5% on MathVista, respectively) with significantly fewer tokens (83.0 and 84.5 tokens). This trend is even more pronounced on MathVision, while baseline models using GRPO need 349.7–440.1 tokens to reach 21%–23% accuracy, our LIPO models hit 23.7% accuracy with only 114.9–120.2 tokens and the token cost is less than one-third of that of the former. Compared with existing R1 mathematical reasoning models, R1-VL-2B (Zhang et al., 2025a) and VLM-R1-3B (Shen et al., 2025), our models achieve better reasoning performance using only one-third of the token count. Figure 1(b) presents the average response length of GRPO (Guo et al., 2025), DAPO (Yu et al., 2025), and our proposed LIPO during the RL training process on the training set. As observed from the figure, the number of tokens used by LIPO gradually decreases throughout the training process, and it achieves a significantly lower token count compared to both GRPO and DAPO methods. This indicates that LIPO effectively optimizes reasoning efficiency by focusing on critical logical steps rather than redundant processes, shaking the inherent notion that higher accuracy necessitates longer responses.

Table 3: Accuracy and average reasoning response token number of models and methods on different Mathematica benchmarks. (Acc ↑, Avg. Token Length ↓)

| Model | Method | MathVista | | MathVision | |
|---|---|---|---|---|---|
| | | Acc (%)↑ | Avg. Token↓ | Acc (%)↑ | Avg. Token↓ |
| Qwen2.5-VL-3B-Instruct | - | 62.3 | 87.5 | 21.2 | 466.5 |
| Qwen2.5-VL-3B-Instruct(baseline) | GRPO | 66.1 | 138.1 | 23.0 | 440.1 |
| R1-VL-2B (Zhang et al., 2025a) | GRPO | 52.1 | 158.8 | 17.1 | 383.2 |
| VLM-R1-3B (Shen et al., 2025) | GRPO | 62.7 | 147.3 | 21.9 | 349.7 |
| **Qwen2.5-VL-3B-Instruct(Ours)** | LIPO | 66.5 | 83.0(↓ 55.1) | 23.3 | 114.9(↓ 325.2) |
| **Tiny-R1V-3B(Ours)** | LIPO | 65.5 | 84.5(↓ 53.6) | 23.7 | 120.2(↓ 319.9) |

**Ablation Study of AMM.** As shown in Table 4, we conduct ablation studies to test the contribution of various designs in AMM, including the design of $\alpha_{i,l}$, $\beta_{i,l}^n$ parameters and gradient projection regularization. Compared with the Wudi Model Merging baseline, $\alpha_{i,l}$, $\beta_{i,l}^n$ parameters can improve the performance by 1.1%. In addition, incorporating the regularization penalty, yields a performance boost of +0.3%. Finally, the AMM model merging achieves the best score of 65.5% on MathVista, reflecting the effectiveness of adaptively adjusting each task vector.

**Hyperparameter studies of LIPO.** To further evaluate the robustness of LIPO, we conduct extensive hyperparameter studies focusing on two critical parameters, the length threshold $L_T$ and the reward threshold $\eta$. Table 5 investigates the impact of $L_T$ on model performance across the Math-Vista and MathVision datasets. Performance peaks at $L_T = 120$ with 66.5% and 23.3% on the two datasets, respectively, before gradually declining as $L_T$ increases further. This trend indicates that $L_T = 120$ is sufficient to achieve best mathematical reasoning performance, and it also proves that LIPO is not sensitive to length restrictions and is effective within a suitable range. Table 5 analyzes the influence of the confidence threshold $\eta$. This shows that a moderate threshold $\eta = 0.2$ can effectively filter responses with similar rewards.

Table 4: Ablation study of Adaptive Model Merging. We study the impact of $\alpha_{i,l}, \beta_{i,l}^n, \mathcal{R}_l$ in AMM.

| Model | MathVista | ChartQA | OCRBench |
|---|---|---|---|
| Qwen2.5-VL-3B-Instruct | 62.3 | 84.0 | 79.7 |
| Math Model with LIPO | 66.5 | 83.9 | 73.5 |
| Structure Model with LIPO | 61.9 | 85.8 | 77.6 |
| OCR Model with LIPO | 57.9 | 83.5 | 84.6 |
| Wudi Model Merging | 63.5 | 85.0 | 81.5 |
| + $\alpha_{i,l}$ | 64.9 | 85.2 | 81.7 |
| + $\beta_{i,l}^n$ | 65.2 | 85.3 | 82.3 |
| + $\mathcal{R}_l$ & Regularized update | 65.5 | 85.3 | 82.5 |

Table 5: The study on the $\eta, L_T$. We study the impact of $\eta, L_T$ in LIPO.

| $L_T$ | MathVista | MathVision |
|---|---|---|
| 100 | 66.1 | 22.9 |
| **120** | **66.5** | **23.3** |
| 140 | 66.2 | 23.1 |
| 160 | 66.1 | 23.0 |

| $\eta$ | MathVista | MathVision |
|---|---|---|
| 0.1 | 66.2 | 23.1 |
| **0.2** | **66.5** | **23.3** |
| 0.3 | 66.1 | 22.8 |
| 0.4 | 66.0 | 22.8 |

## 4.4 QUALITATIVE RESULTS

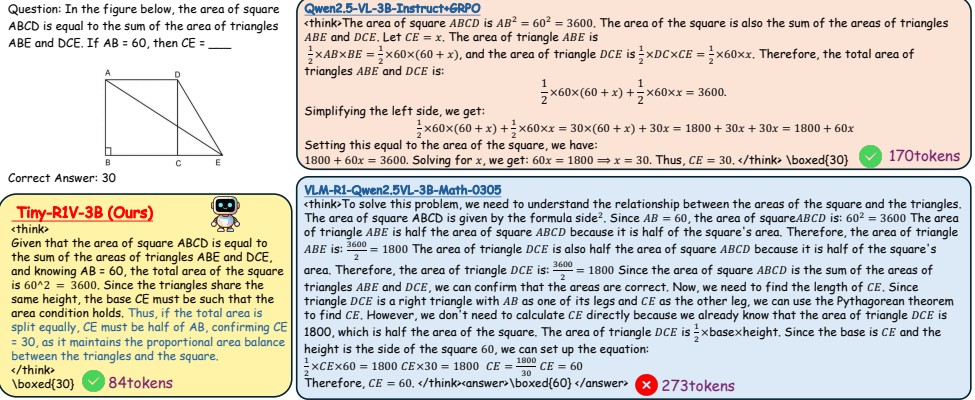

Figure 3: Qualitative Results of reasoning capability of Qwen2.5-VL-3B-Instruct (Bai et al., 2025) with GRPO, VLM-R1-Qwen2.5VL-3B-Math-0305 (official open-source weights) (Shen et al., 2025) and Tiny-R1V-3B (Ours) on mathematical problems.

Figure 3 demonstrates that Tiny-R1V significantly boosts the model's reasoning capabilities when tackling complex mathematical problems. In this example, the model accurately parses the question and obtains the correct answer, showing strong performance in symbolic reasoning and function analysis. Meanwhile, Tiny-R1V uses fewer tokens and simpler thinking methods to obtain the correct results. This proves the superiority of Tiny-R1V in performing lightweight reasoning when handling complex reasoning tasks.

## 5 CONCLUSION

In this paper, we propose Tiny-R1V, a novel lightweight 3B model that unifies multi-task, multi-modal reasoning capabilities while achieving fewer tokens and higher accuracy. Firstly, Tiny-R1V uses LIPO to adjust the proportion of in-group advantage according to the length of the response, and trains specialized reasoning models with shorter responses. Then, it uses AMM to adaptively adjust the weights of task vectors in each model to integrate multiple specialized models into a unified architecture. We conduct extensive experiments and ablation studies, and the result demonstrate the superiority of our proposed Tiny-R1V on various reasoning benchmarks.

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

APPENDIX

## A    THE USE OF LARGE LANGUAGE MODELS (LLMS)

In this paper, large language model (LLM) (Hurst et al., 2024) is only utilized to assist with the refinement of the English language. No LLMs are employed to generate new innovative ideas, and the research process is conducted by the researchers (humans). It is worth noting that the robot image in Figure 1(a) is generated by the Doubao (Gao et al., 2025) image generation foundation model.

## B    BENCHMARKS

Our model is evaluated across four key categories, multimodal mathematical reasoning, multimodal structured data reasoning, OCR capabilities and general capabilities.

### B.1    MATHEMATICS BENCHMARKS

- **MathVista** (Lu et al., 2023) encompasses 6,141 questions spanning diverse domains including arithmetic, geometry, algebra, and statistics.

- **MathVision** (Wang et al., 2024a) is a curated collection of 3,040 high-quality mathematical problems derived from real-world mathematics competitions.

- **MathVerse** (Zhang et al., 2024) comprises 2,612 multimodal mathematical problems, accompanied by 15,672 manually annotated test samples. These samples are categorized into 3 primary question types and 12 subcategories, such as plane geometry, solid geometry, and functions.

- **WeMath** (Qiao et al., 2024) comprises 6,500 visual mathematical problems, covering 67 hierarchical knowledge concepts and five levels of knowledge granularity. Complex problems are decomposed into a novel four-dimensional metric to hierarchically evaluate the inherent issues in the reasoning process of MLLMs.

### B.2    MULTIMODAL STRUCTURED DATA REASONING BENCHMARKS

- **ChartQA**  (Masry et al., 2022) is a large-scale benchmark encompassing 9.6K human-authored questions alongside 23.1K questions generated from human-written chart summaries.

- **CharXiv** (Wang et al., 2024b) involves 2,323 natural, challenging, and diverse charts sourced from scientific papers. It includes both descriptive questions that assess basic chart elements and reasoning questions that require synthesizing information from complex visual elements within the charts.

### B.3    OCR BENCHMARKS

- **OCRbench** (Liu et al., 2024) comprises 1,000 question-answer pairs, with all answers undergoing manual verification and correction, designed to evaluate the OCR capabilities of MLLMs. It encompasses five components: text recognition, scene text-centered VQA, document-oriented VQA, key information extraction, and handwritten mathematical expression recognition.

- **OCR-Reasoning**  (Huang et al., 2025a) consists of 1,069 manually annotated examples, covering 6 core reasoning capabilities and 18 practical reasoning tasks within text-rich visual scenarios. It is specifically designed to systematically evaluate the performance of MLLMs in reasoning tasks involving text-rich images.

### B.4    GENERAL BENCHMARKS

- **MME** (Fu et al., 2023) encompasses 14 subtasks, which are designed to measure the perceptual and cognitive capabilities of MLLMs.

- **MMStar** (Chen et al., 2024a) consists of 1,500 challenging samples meticulously curated by humans, encompassing 6 core functionalities and 18 detailed tasks, aiming to evaluate the multi-modal capacities of MLLMs with a carefully balanced and purified selection of samples.

## C  PROMPTS

> Prompt Template used for Length-Informed Relative Policy Optimization (LIPO).
>
> You FIRST think about the reasoning process as an internal monologue and then provide the final answer. The reasoning process MUST BE enclosed within </think> </think> tags. The final answer MUST BE put in \boxed{}.

## D  INFERENCE TIME

We compare the inference time across different benchmarks on eight A800 40G GPUs. As illustrated in Table 6, Tiny-R1V-3B (with LIPO) exhibits notable inference speed superiority over Qwen2.5-VL-3B-Instruct (with GRPO) across all evaluated benchmarks. Tiny-R1V employs fewer tokens during the inference process, resulting in significantly faster inference on each benchmark.

Table 6: Inference time on different benchmarks

| Model | MathVista | ChartQA | OCRBench |
|---|---|---|---|
| Qwen2.5-VL-3B-Instruct(GRPO) | 1788.4s | 977.2s | 788.4s |
| Tiny-R1V-3B(LIPO,Ours) | 623.4s | 455.4s | 318.6s |

## E  MATHEMATICS REASONING CAPABILITIES CASE STUDIES

This section demonstrates the significant enhancement in reasoning capabilities of Tiny-R1V when addressing complex mathematical reasoning tasks. In these case studies, Tiny-R1V-3B accurately parses problems and derives correct answers, showcasing its robust proficiency in symbolic reasoning, geometric comprehension, and mathematical computation. Meanwhile, Tiny-R1V-3B achieves correct results using fewer tokens and simpler reasoning approaches, which demonstrate the superiority of its lightweight reasoning in handling complex reasoning tasks. Specifically, mathematical reasoning tasks require rigorous chain-of-mathematical-thought reasoning. Existing methods (such as Qwen2.5-VL-3B-Instruct (Bai et al., 2025), Qwen2.5VL-3B-GRPO (Shao et al., 2024), VLM-R1-Qwen2.5VL-3B-Math-0305 (Shen et al., 2025) ) tend to employ multi-step complex computations and numerous equation transformation operations when tackling complex reasoning tasks. In contrast, our Tiny-R1V-3B can precisely leverage existing mathematical theorems to simplify the mathematical derivation process (see Case 1 and Case 2 for details). This simplified derivation process substantially reduces the token consumption during reasoning while ensuring the correctness of the reasoning process.

> **Mathematics Reasoning Capabilities Case1**
>
> 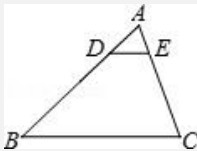
>
> **Question:** As shown in the figure, $D$ and $E$ are the points on the edges $AB$ and $AC$ of $\triangle ABC$, $DE \parallel BC$, if $AD : DB = 1.0 : 3.0$, $AE = 2.0$, then the length of $AC$ is (   ).

**Qwen2.5-VL-3B-Instruct Response**: To solve for the length of $AC$, we start by analyzing the given information and the geometric relationships in the triangle.

1. **Identify the coordinates of points $A$, $B$, and $C$:**

   - Let $A = (0, 4)$, $B = (0, 0)$, and $C = (4, 0)$.
   - Since $AD : DB = 1 : 3$, point $D$ divides $AB$ in the ratio 1:3. Therefore, the coordinates of $D$ are $\left(0, \frac{4 \cdot 1}{1+3}\right) = (0, 1)$.
   - Since $DE \parallel BC$, point $E$ divides $AC$ in the same ratio as $D$ divides $AB$, which is 1:3. Therefore, the coordinates of $E$ are $\left(\frac{4 \cdot 1}{1+3}, 0\right) = (1, 0)$.

2. **Calculate the length of $AC$:**

   - he length of $AC$ is the distance from $A = (0, 4)$ to $C = (4, 0)$. Using the distance formula:
     $$AC = \sqrt{(4-0)^2 + (0-4)^2} = \sqrt{4^2 + (-4)^2} = \sqrt{16 + 16} = \sqrt{32} = 4\sqrt{2}$$

Thus, the length of $AC$ is $\boxed{4\sqrt{2}}$.     **[Incorrect]**

---

**Qwen2.5VL-3B-GRPO Response**: `<think>` Since $DE \parallel BC$, triangles $ADE$ and $ABC$ are similar by the AA similarity criterion. Therefore, the ratios of corresponding sides are equal. Given $AD : DB = 1 : 3$, we can infer that $AD = \frac{1}{4}AB$ and $DB = \frac{3}{4}AB$. Since $AB = AD + DB = 4$ cm, we have $AD = 1$ cm and $DB = 3$ cm. Because $DE \parallel BC$, the ratio $AD : AB = AE : AC$. Given $AE = 2$ cm, we can set up the proportion:

$$\frac{AD}{AB} = \frac{AE}{AC} \implies \frac{1}{4} = \frac{2}{AC} \implies AC = 8 \text{ cm}$$

Thus, the length of $AC$ is 8 cm. `</think>` $\boxed{8}$     **[Correct]**

---

**VLM-R1-Qwen2.5VL-3B-Math-0305**: `<think>` 1. Given that $AD : DB = 1 : 3$, we can determine the length of $AD$ and $DB$. 2. Since $AD + DB = AB$ and $AB = 4$, we can write:
$$AD + DB = 4$$
3. Given $AD : DB = 1 : 3$, we can express $AD$ and $DB$ in terms of a common variable $x$:
$$AD = x \quad \text{and} \quad DB = 3x$$
4. Substituting these into the equation $AD + DB = 4$:
$$x + 3x = 4$$
5. Solving for $x$:
$$4x = 4 \implies x = 1$$
6. Therefore, $AD = 1$ and $DB = 3$.
Since $DE \parallel BC$, $DE$ is parallel to $BC$ and $DE$ is half the length of $BC$ because $AD : DB = 1 : 3$.
7. Given $AE = 2$, we can determine the length of $AC$:
$$AC = AE + EC$$
8. Since $DE \parallel BC$, $EC = DB = 3$. 9. Therefore:
$$AC = AE + EC = 2 + 3 = 5$$
`</think> <answer>`5`</answer>`     **[Incorrect]**

---

**Tiny-R1V-3B Response (Ours):** `<think>` Given that $DE \parallel BC$, it implies that $\triangle ADE \sim \triangle ABC$ by the Basic Proportionality Theorem (or Thales' theorem). Therefore, the ratios of corresponding sides are equal. Since $AD : DB = 1 : 3$, $AD : AB = 1 : 4$. Thus, $AE : AC = 1 : 4$. Given $AE = 2$, we can find $AC$ as $AC = 4 \times AE = 4 \times 2 = 8$. `</think>`
Answer: $\boxed{8}$     **[Correct]**

**Mathematics Reasoning Capabilities Case2**

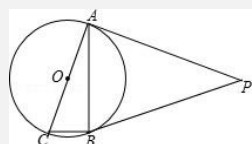

**Question:** As shown in the figure, $PA$ and $PB$ are tangents of $\odot O$, with tangent points $A$ and $B$, $AC$ is the diameter of $\odot O$, given that $\angle P = 50.0$, then the size of $\angle ACB$ is (    ).

- - - - - - - - - - - - - - - - - - - - - - - - - - - - - - - - - - - - - - - - - - - - -

**Qwen2.5-VL-3B-Instruct Response**: To solve for the size of $\angle ACB$, we start by analyzing the given information and the geometric properties of the circle and the tangents.

1. **Identify the properties of the tangents and the circle:**

    - $PA$ and $PB$ are tangents to the circle $\odot O$ at points $A$ and $B$ respectively.

    - $AC$ is the diameter of the circle.

    - $\angle P = 50°$.

2. **Use the property of tangents:**

    - The tangents drawn from an external point to a circle are equal in length. Therefore, $PA = PB$.

    - The line segment $OA$ is perpendicular to the tangent $PA$ at the point of tangency $A$, so $\angle OAP = 90°$.

    - Similarly, $\angle OBP = 90°$.

3. **Determine the angles in $\triangle OAP$ and $\triangle OBP$:**

    - Since $\angle OAP = 90°$ and $\angle OBP = 90°$, the sum of the angles in $\triangle OAP$ and $\triangle OBP$ is $180°$.

    - Therefore, $\angle AOP = 180° - \angle P = 180° - 50° = 130°$.

4. **Use the fact that $AC$ is the diameter:**

    - Since $AC$ is the diameter, $\angle ABC = 90°$ (angle in a semicircle).

    - Now, consider $\triangle OAC$. Since $OA = OB$ (both are radii of the circle), $\triangle OAC$ is isosceles with $\angle OAC = \angle OBA$.

    - The sum of the angles in $\triangle OAC$ is $180°$. Since $\angle OAC = \angle OBA$ and $\angle OAP = 90°$, we have $\angle OAC = \angle OBA = \frac{180° - 90°}{2} = 45°$.

    - Therefore, $\angle ACB = \angle OAC + \angle OBA = 45° + 45° = 90° - \angle P/2 = 90° - 25° = 65°$.

Thus, the size of $\angle ACB$ is $\boxed{65°}$.        **[Correct]**

- - - - - - - - - - - - - - - - - - - - - - - - - - - - - - - - - - - - - - - - - - - - -

**Qwen2.5VL-3B-GRPO Response**: `<think>`
Since $PA$ and $PB$ are tangents to $\odot O$ from point $P$, $PA = PB$. Therefore, $\triangle PAB$ is isosceles with $\angle P = 50°$. The sum of the angles in a triangle is $180°$, so we have:

$$\angle PAB = \angle PBA = \frac{180° - 50°}{2} = 65°.$$

Since $AC$ is a diameter, $\angle ABC = 90°$ (angle in a semicircle). Now, $\angle ACB$ can be found by noting that $\angle ACB = 180° - \angle CAB - \angle ABC$. Since $\angle CAB = \angle PAB = 65°$, we have:

$$\angle ACB = 180° - 65° - 90° = 25°.$$

`</think>` $\boxed{25°}$      **[Incorrect]**

- - - - - - - - - - - - - - - - - - - - - - - - - - - - - - - - - - - - - - - - - - - - -

**VLM-R1-Qwen2.5VL-3B-Math-0305**: `<think>` To solve for the size of $\angle ACB$, we start by noting that $PA$ and $PB$ are tangents to the circle $\odot O$ at points $A$ and $B$ respectively.

This means that $OA$ and $OB$ are perpendicular to $PA$ and $PB$ respectively. Therefore, $\angle OAP = 90°$ and $\angle OBP = 90°$.

Since $AC$ is the diameter of the circle, $\angle ABC = 90°$ because any angle inscribed in a semicircle is a right angle. Now, we need to find $\angle ACB$. We know that $\angle P = 50°$, and since $PA$ and $PB$ are tangents, $\angle OAP = 90°$ and $\angle OBP = 90°$. The angle $\angle APB$ is the external angle to $\triangle OAP$ and $\triangle OBP$, and it is equal to the sum of the opposite internal angles, which are $\angle OAP$ and $\angle OBP$. Therefore, $\angle APB = 180° - \angle OAP - \angle OBP = 180° - 90° - 90° = 180° - 180° + 50° = 50°$.

Now, we can use the fact that the sum of the angles in $\triangle APB$ is $180°$. So, $\angle BAP + \angle ABP + \angle APB = 180°$. Since $\angle BAP = \angle OAP = 90°$ and $\angle ABP = \angle OBP = 90°$, we have $\angle BAP + \angle ABP = 180° - \angle APB = 180° - 50° = 130°$. Therefore, $\angle ACB = 180° - \angle ABC - \angle BAC = 180° - 90° - 40° = 50°$.

Thus, the size of $\angle ACB$ is $\boxed{40°}$. `</think>`

`<answer>` 40° `</answer>`      **[Incorrect]**

- - - - - - - - - - - - - - - - - - - - - - - - - - - - - - - - - - - - - - - - - - - - -

**Tiny-R1V-3B Response (Ours):** `<think>` Since $PA$ and $PB$ are tangents, $\angle OAP$ and $\angle OBP$ are both $90°$. Thus, $\angle AOB$ is supplementary to $\angle P$, making it $130°$. Since $AC$ is the diameter, $\angle ACB$ is an inscribed angle subtending the same arc, thus $\angle ACB$ is half of the central angle, which is $65°$. Therefore, $\angle ACB$ is half of the remaining angle at the circumference. `</think>`

Answer: $\boxed{65°}$      **[Correct]**

---

**Mathematics Reasoning Capabilities Case3**

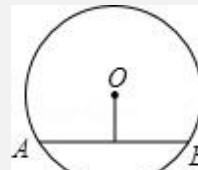

**Question:** As shown in the figure, in $\odot O$ with a radius of 5.0, the length of the chord $AB$ is 8.0, then the distance from the center $O$ to the chord $AB$ is (    ).

- - - - - - - - - - - - - - - - - - - - - - - - - - - - - - - - - - - - - - - - - - - - -

**Qwen2.5-VL-3B-Instruct Response**: To find the distance from the center $O$ to the chord $AB$ in the circle, we can use the Pythagorean theorem. Here's the step-by-step solution:

1. **Identify the given information:**
   - The radius of the circle $OA = OB = 5.0$.
   - The length of the chord $AB = 8.0$.

2. **Draw the perpendicular from the center $O$ to the chord $AB$:**
   - Let the perpendicular from $O$ to $AB$ intersect $AB$ at point $M$. Since $OM$ is perpendicular to $AB$, $M$ is the midpoint of $AB$.
   - Therefore, $AM = MB = \frac{AB}{2} = \frac{8.0}{2} = 4.0$.

3. **Form a right triangle $OAM$:**
   - In the right triangle $OAM$, $OA$ is the hypotenuse, $AM$ is one leg, and $OM$ is the other leg.
   - We know $OA = 5.0$ and $AM = 4.0$.
   - We need to find the length of $OM$.

4. **Apply the Pythagorean theorem:**
$$OA^2 = OM^2 + AM^2$$

Substituting the known values:
$$5.0^2 = OM^2 + 4.0^2$$

$$25.0 = OM^2 + 16.0$$
$$OM^2 = 25.0 - 16.0$$
$$OM^2 = 9.0$$
$$OM = \sqrt{9.0} = 3.0$$

Therefore, the distance from the center $O$ to the chord $AB$ is $\boxed{3.0}$.  **[Correct]**

---

**Qwen2.5VL-3B-GRPO Response**: `<think>` To find the distance from the center $O$ to the chord $AB$, we can use the geometric properties of the circle. Specifically, we can use the fact that the perpendicular from the center of the circle to a chord bisects the chord. Let's denote the distance from the center $O$ to the chord $AB$ as $d$. The perpendicular from $O$ to $AB$ will split $AB$ into two equal segments, each of length 4 (since $AB = 8$).

We can now consider the right triangle $OAC$ where $OC$ is the distance from the center to the chord, $OA$ is the radius of the circle, and $AC$ is half the length of the chord $AB$. By the Pythagorean theorem, we have:

$$OA^2 = OC^2 + AC^2$$

Substituting the known values $OA = 5$ and $AC = 4$, we get:

$$5^2 = d^2 + 4^2$$
$$25 = d^2 + 16$$
$$d^2 = 25 - 16$$
$$d^2 = 9$$
$$d = \sqrt{9} = 3$$

So, the distance from the center $O$ to the chord $AB$ is 3. `</think>` $\boxed{3}$  **[Correct]**

---

**VLM-R1-Qwen2.5VL-3B-Math-0305**: `<think>` To find the distance from the center $O$ to the chord $AB$, we can use the Pythagorean theorem. Here's the step-by-step reasoning:

1. **Identify the given information:**

   - The radius of the circle $\odot O$ is 5.0.

   - The length of the chord $AB$ is 8.0.

2. **Draw the perpendicular from the center $O$ to the chord $AB$:**

   - Let the perpendicular from $O$ to $AB$ intersect $AB$ at point $M$. Since $OM$ is perpendicular to $AB$, $M$ is the midpoint of $AB$.

   - Therefore, $AM = MB = \frac{AB}{2} = \frac{8.0}{2} = 4.0$.

3. **Form a right triangle $OMA$:**

   - In the right triangle $OMA$, $OA$ is the radius of the circle, so $OA = 5.0$.

   - $AM$ is half the length of the chord $AB$, so $AM = 4.0$.

   - $OM$ is the distance from the center $O$ to the chord $AB$, which we need to find.

4. **Apply the Pythagorean theorem:**

   - The Pythagorean theorem states that in a right triangle, the square of the hypotenuse (the side opposite the right angle) is equal to the sum of the squares of the other two sides.

   - In triangle $OMA$, $OA$ is the hypotenuse, $OM$ is one leg, and $AM$ is the other leg.

   - Therefore, $OA^2 = OM^2 + AM^2$.

   - Substituting the known values, we get $5.0^2 = OM^2 + 4.0^2$.

   - Simplifying, we get $25.0 = OM^2 + 16.0$.

   - Solving for $OM^2$, we get $OM^2 = 25.0 - 16.0 = 9.0$.

- Taking the square root of both sides, we get $OM = \sqrt{9.0} = 3.0$.

Therefore, the distance from the center $O$ to the chord $AB$ is 3.0.
`</think> <answer>3.0</answer>`    **[Correct]**

---

**Tiny-R1V-3B Response (Ours):** `<think>`Given a circle with radius 5 and chord $AB = 8$, we can use the right triangle formed by the radius, the perpendicular from the center to the chord, and half of the chord. The perpendicular bisects $AB$ into two segments of 4 each, so we use the Pythagorean theorem: $5^2 = 4^2 + d^2$, where $d$ is the distance from $O$ to $AB$. Thus, $25 = 16 + d^2$, so $d^2 = 9$ and $d = 3$. `</think>`
Answer: $\boxed{3}$    **[Correct]**

## F    TABLE, CHART AND DOCUMENT REASONING CAPABILITIES CASE STUDIES

This section demonstrates a significant improvement in the reasoning capabilities of Tiny-R1V when tackling table, chart, and document reasoning tasks. In these case studies, Tiny-R1V-3B can accurately parse questions and derive correct answers. Structured data reasoning tasks require both perception and reasoning capabilities; however, the perception process may consume a large number of redundant tokens. As the cases below, Tiny-R1V-3B has reduced such redundant perception processes, thereby achieving a more efficient perception process and a more accurate reasoning process.

---

**Table, Chart and Document Reasoning Capabilities Capabilities Case1**

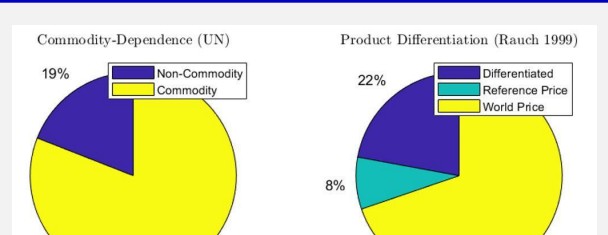

**Question:** What is the name of the chart with the highest percentage for its largest category?

---

**Qwen2.5-VL-3B-Instruct Response**: The chart with the highest percentage for its largest category is the one on the right, which is labeled "Product Differentiation (Rauch 1999)".
**[Incorrect]**

---

**Qwen2.5VL-3B-GRPO Response**: `<think>` Looking at the two pie charts, the left one shows "Commodity-Dependence (UN)" with 81% for Commodity and 19% for Non-Commodity. The right one shows "Product Differentiation (Rauch 1999)" with 70% for World Price, 22% for Differentiated, and 8% for Reference Price. The chart with the highest percentage for its largest category is the left one, as it has 81% for Commodity. `</think>`
Commodity-Dependence (UN)    **[Correct]**

---

**Tiny-R1V-3B Response (Ours):** `<think>` The chart on the left shows a clear majority of the category labeled "Commodity" at 81%, indicating it's the largest segment. The right chart shows a significant portion for "World Price" at 70%, but the left chart's largest category is more dominant. `</think>`

Answer: Commodity-Dependence (UN)    **[Correct]**

---

**Table, Chart and Document Reasoning Capabilities Capabilities Case2**

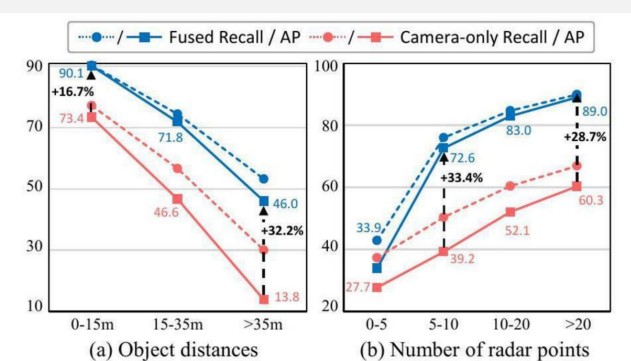

(a) Object distances    (b) Number of radar points

**Question:** Does increase the number of radar points close the gap between Fused recall and Camera-only recall?

---

**Qwen2.5-VL-3B-Instruct Response**: Yes, increasing the number of radar points generally closes the gap between Fused Recall and Camera-only Recall. This can be seen in both graphs:

1. **Graph (a): Object Distances**

   - The Fused Recall/AP line (solid blue) shows a significant improvement as the object distance increases.

   - The Camera-only Recall/AP line (dashed red) shows a corresponding decrease in performance.

   - As the object distance increases, the gap between the two lines narrows, indicating that the fused approach performs better.

2. **Graph (b): Number of Radar Points**

   - The Fused Recall/AP line (solid blue) shows a significant improvement as the number of radar points increases.

   - The Camera-only Recall/AP line (dashed red) shows a corresponding decrease in performance.

   - As the number of radar points increases, the gap between the two lines narrows, indicating that the fused approach performs better.

In both cases, the fused approach (solid lines) consistently outperforms the camera-only approach (dashed lines) as the number of radar points or object distances increase.
**[Correct]**

---

**Qwen2.5VL-3B-GRPO Response**: `<think>` The graph shows two lines: one for "Fused Recall / AP" (solid blue line) and one for "Camera-only Recall / AP" (dotted red line). The solid blue line starts at 90.1% for 0-15m and decreases to 46.0% for >35m. The dotted red line starts at 73.4% for 0-15m and decreases to 13.8% for >35m. The gap between the two lines is significant, especially at shorter distances. As the number of radar points increases (from 0-5 to >20), the solid blue line (Fused Recall) increases from 33.9% to 89.0%, while the dotted red line (Camera-only Recall) increases from 27.7% to 60.3%. This indicates that increasing the number of radar points does indeed help close the gap between the Fused Recall and Camera-only Recall. `</think>` Yes    **[Correct]**

---

**Tiny-R1V-3B Response (Ours):** `<think>` The graphs show that as the number of radar points increases, the Fused Recall/AP line approaches the Camera-only Recall/AP line, indicating a closing gap. This suggests that more radar data points improve the recall and AP metrics, aligning the performance closer to the camera-only results. The percentage increases also support this, showing improvements as the number of points grows. `</think>`

Answer: | Yes |        **[Correct]**

---

**Table, Chart and Document Reasoning Capabilities Capabilities Case3**

**ITEM 6. SELECTED FINANCIAL DATA.**

The following selected financial data has been derived from our audited financial statements. This data should be read in conjunction with Item 7— "Management's Discussion and Analysis of Financial Condition and Results of Operations" and our financial statements and related notes thereto included elsewhere in this Form 10-K. Our historical results are not necessarily indicative of operating results to be expected in the future.

**Selected Financial Data**
*(in thousands, except per share data)*

| | | 2019 | | 2018 | | 2017 | | 2016 | | 2015 |
|---|---|---|---|---|---|---|---|---|---|---|
| **Income Statement Data** | | | | | | | | | | |
| Revenue | $ | 84,590 | $ | 63,559 | $ | 45,390 | $ | 34,701 | $ | 25,367 |
| Operating income (loss) | $ | (4,590) | $ | (7,806) | $ | 2,769 | $ | 1,824 | $ | 1,892 |
| Net income (loss) | $ | (724) | $ | (11,807) | $ | 14,092 | $ | 1,959 | $ | 2,526 |
| Net income (loss) per share—basic | $ | (0.02) | $ | (0.33) | $ | 0.43 | $ | 0.06 | $ | 0.08 |
| Net income (loss) per share—diluted | $ | (0.02) | $ | (0.33) | $ | 0.40 | $ | 0.06 | $ | 0.08 |
| **Balance Sheet Data** | | | | | | | | | | |
| Working capital | $ | 34,082 | $ | 17,221 | $ | 41,342 | $ | 31,980 | $ | 24,005 |
| Total assets | $ | 135,897 | $ | 127,150 | $ | 71,719 | $ | 48,385 | $ | 38,746 |
| Other borrowings | $ | 556 | $ | 810 | $ | — | $ | — | $ | — |
| Stockholders' equity | $ | 107,333 | $ | 95,394 | $ | 61,408 | $ | 39,485 | $ | 30,433 |

**Question:** What is the percentage change in working capital from 2018 to 2019?

---

**Qwen2.5-VL-3B-Instruct Response**: To calculate the percentage change in working capital from 2018 to 2019, we use the formula:

$$\text{Percentage Change} = \left( \frac{\text{New Value} - \text{Old Value}}{\text{Old Value}} \right) \times 100$$

Given:

- Working capital in 2019: $34,082$
- Working capital in 2018: $17,221$

Plugging in the values:

$$\text{Percentage Change} = \left( \frac{34,082 - 17,221}{17,221} \right) \times 100$$

$$\text{Percentage Change} = \left( \frac{16,861}{17,221} \right) \times 100$$

$$\text{Percentage Change} \approx 97.48\%$$

So, the percentage change in working capital from 2018 to 2019 is approximately 97.48%. **[Incorrect]**

---

**Qwen2.5VL-3B-GRPO Response**: `<think>` To calculate the percentage change in working capital from 2018 to 2019: Percentage change = (New value - Original value) / Original value * 100

Working capital in 2018 = $17,221$
Working capital in 2019 = $34,082$
Percentage change = ($34,082$ - $17,221$) / $17,221$ * 100
Percentage change = $16,861$ / $17,221$ * 100
Percentage change  97.28%
`</think>` | 97.28% |        **[Incorrect]**

**Tiny-R1V-3B Response (Ours):** `<think>` The working capital in 2018 was $17,221 thousand and in 2019 it was $34,082 thousand. The change is calculated as (34,082 - 17,221) / 17,221 * 100 = 97.9%. This indicates an increase of approximately 97.9% in working capital from 2018 to 2019.`</think>`

Answer: 97.9%        **[Correct]**

**Table, Chart and Document Reasoning Capabilities Capabilities Case4**

**2019 Annual Report**
Management's Discussion and Analysis of Financial Condition and Results of Operations *continued*

**Consolidated Revenues**

|  |  |  | (dollars in millions) Increase/(Decrease) | |
|---|---|---|---|---|
| Years Ended December 31, | **2019** | 2018 | **2019 vs. 2018** | |
| Consumer | $ **91,056** | $ 89,762 | $ 1,294 | 1.4% |
| Business | **31,443** | 31,534 | (91) | (0.3) |
| Corporate and other | **9,812** | 9,936 | (124) | (1.2) |
| Eliminations | **(443)** | (369) | (74) | 20.1 |
| **Consolidated Revenues** | $ **131,868** | $ 130,863 | $ 1,005 | 0.8 |

Consolidated revenues increased $1.0 billion, or 0.8%, during 2019 compared to 2018, primarily due to an increase in revenues at our Consumer segment, partially offset by decreases in revenues at our Business segment and Corporate and other.

Revenues for our segments are discussed separately below under the heading "Segment Results of Operations."

Corporate and other revenues decreased $124 million, or 1.2%, during 2019 compared to 2018, primarily due to a decrease of $232 million in revenues within Verizon Media.

**Consolidated Operating Expenses**

|  |  |  | (dollars in millions) Increase/(Decrease) | |
|---|---|---|---|---|
| Years Ended December 31, | **2019** | 2018 | **2019 vs. 2018** | |
| Cost of services | $ **31,772** | $ 32,185 | $ (413) | (1.3)% |
| Cost of wireless equipment | **22,954** | 23,323 | (369) | (1.6) |
| Selling, general and administrative expense | **29,896** | 31,083 | (1,187) | (3.8) |
| Depreciation and amortization expense | **16,682** | 17,403 | (721) | (4.1) |
| Media goodwill impairment | **186** | 4,591 | (4,405) | (95.9) |
| **Consolidated Operating Expenses** | $ **101,490** | $ 108,585 | $ (7,095) | (6.5) |

Operating expenses for our segments are discussed separately below under the heading "Segment Results of Operations."

**Cost of Services**

Cost of services includes the following costs directly attributable to a service: salaries and wages, benefits, materials and supplies, content costs, contracted services, network access and transport costs, customer provisioning costs, computer systems support, and costs to support our outsourcing contracts and technical facilities. Aggregate customer care costs, which include billing and service provisioning, are allocated between Cost of services and Selling, general and administrative expense.

Cost of services decreased $413 million, or 1.3%, during 2019 compared to 2018, primarily due to decreases in network access costs, a product realignment charge in 2018 (see "Special Items"), decreases in employee-related costs resulting from the Voluntary Separation Program and decreases in digital content costs. These decreases were partially offset by increases in rent expense as a result of adding capacity to the networks to support demand and the adoption of the new lease accounting standard in 2019, regulatory fees, and costs related to the device protection package offered to our wireless retail postpaid customers.

**Cost of Wireless Equipment**

Cost of wireless equipment decreased $369 million, or 1.6%, during 2019 compared to 2018, primarily as a result of declines in the number of wireless devices sold as a result of an elongation of the handset upgrade cycle, partially offset by a shift to higher priced devices in the mix of wireless devices sold.

**Selling, General and Administrative Expense**

Selling, general and administrative expense includes salaries and wages and benefits not directly attributable to a service or product, bad debt charges, taxes other than income taxes, advertising and sales commission costs, call center and information technology costs, regulatory fees, professional service fees, and rent and utilities for administrative space. Also included is a portion of the aggregate customer care costs as discussed above in "Cost of Services."

Selling, general and administrative expense decreased $1.2 billion, or 3.8%, during 2019 compared to 2018, primarily due to decreases in employee-related costs primarily due to the Voluntary Separation Program, a decrease in severance, pension and benefits charges (see "Special Items"), the acquisition and integration related charges in 2018 primarily related to the acquisition of

**Verizon Communications Inc. and Subsidiaries** 2019 Annual Report  **11**

**Question:** What was the average corporate segment revenue and other segment revenue for 2018 and 2019?

**Qwen2.5-VL-3B-Instruct Response**: To calculate the average corporate segment revenue and other segment revenue for 2018 and 2019, we need to follow these steps:
1. Identify the revenue for the corporate and other segment for both years. 2. Calculate the average of these two values.
From the table:

- Corporate and other revenue for 2019: $9,812$ million

- Corporate and other revenue for 2018: $9,936$ million

Now, we calculate the average:

$$\text{Average} = \frac{2019 \text{ Revenue} + 2018 \text{ Revenue}}{2}$$

$$\text{Average} = \frac{9,812 + 9,936}{2}$$

$$\text{Average} = \frac{19,748}{2}$$

$$\text{Average} = 9,874 \text{ million dollars}$$

So, the average corporate segment revenue and other segment revenue for 2018 and 2019 is $9,874$ million.    **[Correct]**

---

**Qwen2.5VL-3B-GRPO Response**: `<think>` To find the average corporate and other segment revenue for 2018 and 2019, we need to add the revenues for both years and divide by 2. For 2019, the corporate and other segment revenue is $9,812$ million. For 2018, the corporate and other segment revenue is $9,936$ million. Average = (9,812 + 9,936) / 2 Average = 19,748 / 2 Average = $9,874$ million Therefore, the average corporate and other segment revenue for 2018 and 2019 is $9,874$ million. `</think>` $\boxed{9874}$    **[Correct]**

---

**Tiny-R1V-3B Response (Ours):** `<think>` To find the average, I'll add the revenues for both segments in 2018 and 2019, then divide by 2. For Corporate and other, it's (9,936 + 9,812) / 2 = 9874. `</think>` $\boxed{9874}$    **[Correct]**

## G  OCR CAPABILITIES CASE STUDIES

This section demonstrates the capability of Tiny-R1V in OCR tasks. In these case studies, Tiny-R1V-3B can accurately recognize image content and derive correct answers, and exhibits accurate performance in tasks such as formula recognition, image-based question answering, and character parsing.

---

**OCR Capabilities Case1**

$$\underset{\mathbf{u}}{\arg\min} \ \frac{1}{2}\|\mathbf{u}\|^2 + C \sum_{\forall i,j \mathbf{v_{t_i}} \succ \mathbf{v_{t_j}}} \epsilon_{ij}$$

$$s.t. \ \mathbf{u}^T \cdot (\mathbf{v_{t_i}} - \mathbf{v_{t_j}}) \geq 1 - \epsilon_{ij}$$

$$\epsilon_{ii} \geq 0.$$

**Question:** Please write out the expression of the formula in the image using LaTeX format.

**Tiny-R1V-3B Response:** `<think>`

$$\arg\min_u \frac{1}{2}\|u\|^2 + C \sum_{\forall i,j\mathbf{v}_{ti}>\mathbf{v}_{tj}} \epsilon_{ij}$$
$$\text{s.t. } \mathbf{u}^T \cdot (\mathbf{v}_{ti} - \mathbf{v}_{tj}) \geq 1 - \epsilon_{ij}$$
$$\epsilon_{ij} \geq 0.$$

`</think>`

Answer:
$$\arg\min_u \frac{1}{2}\|u\|^2 + C \sum_{\forall i,j\mathbf{v}_{ti}>\mathbf{v}_{tj}} \epsilon_{ij}$$
$$\text{s.t. } \mathbf{u}^T \cdot (\mathbf{v}_{ti} - \mathbf{v}_{tj}) \geq 1 - \epsilon_{ij}$$
$$\epsilon_{ij} \geq 0.$$

**[Correct]**

---

**OCR Capabilities Case2**

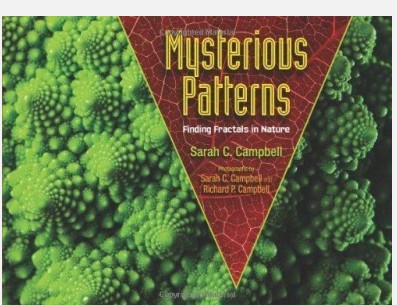

**Question:** Please write out the expression of the formula in the image using LaTeX format.

- - - - - - - - - - - - - - - - - - - - - - - - - - - - - - - - - - - - - - - - - - - - - - - - - -

**Tiny-R1V-3B Response:** `<think>` Sarah C. Campbell `</think>`

Answer: Sarah C. Campbell    **[Correct]**

---

**OCR Capabilities Case3**

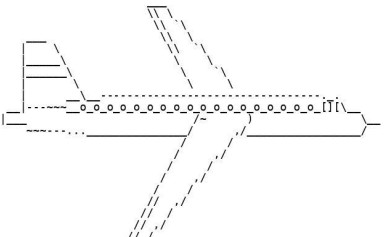

**Question:** Which of the following options best matches the image? Here's some options: submarine, airplane, ship, chopper.

- - - - - - - - - - - - - - - - - - - - - - - - - - - - - - - - - - - - - - - - - - - - - - - - - -

**Tiny-R1V-3B Response:** `<think>` The image depicts a shape that resembles an airplane with wings and a tail. The dashed lines and the overall structure suggest an aircraft. Among the given options, the one that best matches the image is an airplane. `</think>`

Answer: airplane    **[Correct]**

> **OCR Capabilities Case4**
>
> 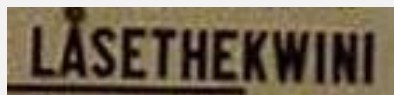
>
> **Question:** How many times does the character 'e' appear in the picture? Please output the exact number without any additional explanation.
>
> - - - - - - - - - - - - - - - - - - - - - - - - - - - - - - - - - - - - - - - - - - - - -
>
> **Tiny-R1V-3B Response:** `<think>` I need to count the occurrences of the character 'e' in the given text. I'll go through each character and count how many times 'e' appears. `</think>`
> Answer: 2   **[Correct]**

## H  LIMITATIONS

Due to resource constraints, our experiments are limited to 3B-parameter models. Future work will explore the integration of MLLMs with larger parameter sizes, multilingual support, and a broader range of reasoning tasks. By incorporating more advanced chain-of-thought techniques such as think with images, we aim to enhance its reasoning capabilities on more complex visual-logical reasoning tasks. Additionally, we plan to filter out low-quality samples using rules during the preprocessing stage. For evaluation purposes, we intend to develop new benchmarks specifically designed to assess the reasoning capabilities of MLLMs.

