# OpenReview forum: "Tiny-R1V: Lightweight Multimodal Unified Reasoning Model via Model Merging"
_ICLR.cc/2026/Conference — ICLR 2026 Conference Withdrawn Submission_

### Official Review · Reviewer_pmBN · 2025-10-31

**Soundness:** 3
**Presentation:** 2
**Contribution:** 2
**Rating:** 4
**Confidence:** 4

**Summary:**

This paper proposes Tiny-R1V, a lightweight multimodal reasoning model built on Qwen2.5-VL-3B. The framework has two main components:
(1) Length-Informed Relative Policy Optimization (LIPO) — a reinforcement learning variant that dynamically reweights shorter yet high-quality responses, encouraging concise reasoning chains;
(2) Adaptive Model Merging (AMM) — a training-free merging strategy that combines multiple task-specialized models (math, structured data, OCR) into a unified model via dual-weighting (α, β) and gradient projection regularization.

**Strengths:**

Novel efficiency-oriented RL design: LIPO elegantly penalizes over-lengthy reasoning without harming accuracy, a meaningful step toward “thinking efficiently” for small MLLMs.

Training-free merging approach: AMM’s adaptive weighting and projection regularization address task interference more robustly than prior WUDI or TIES merging.

Comprehensive evaluation: Covers math, structure, OCR, and general-ability benchmarks, showing consistent improvements and qualitative evidence of shorter, correct reasoning traces.

**Weaknesses:**

Unclear multimodal specificity: LIPO and AMM appear largely modality-agnostic. The framework seems equally applicable to text-only reasoning without visual modifications, which may weaken the “multimodal” positioning.

Marginal gains over mixture training: The improvements over joint mixture training are small, calling into question the practical advantage of the proposed pipeline.

**Questions:**

Mixture-training vs. merging: Is the “All(Mix)” dataset (35K) exactly the union of task sets (Math 15K + Structure 15K + OCR 10K = 40K)? If not, why do sizes differ, and could that confound conclusions.

Text-only applicability: Has your method (LIPO and/or AMM) been proposed in text-only settings? If not, please clarify whether the approach is modality-agnostic, and specify what (if anything) would need to change to make it work for purely textual reasoning tasks.

Table 3 clarification: In Table 3, Qwen2.5-VL-3B-Instruct (Ours) + LIPO and Tiny-R1V-3B (Ours) show nearly identical accuracy and token lengths. What concretely differs between these two models?

---

### Official Review · Reviewer_NgFq · 2025-10-31

**Soundness:** 2
**Presentation:** 1
**Contribution:** 2
**Rating:** 2
**Confidence:** 3

**Summary:**

The authors propose to improve the reasoning capabilities of lightweight MLLMs to achieve faster inference and higher performance. To this end, they introduce LIPO, an extension of GRPO that not only optimizes for maximum rewards but also constrains response length. The goal is to find a Pareto-optimal solution that balances response quality and conciseness. After obtaining lightweight MLLMs optimized for each task, the authors further propose merging them using AMM, a training-free method that adjusts task vectors (the differences between task-specific and base model weights) to minimize conflicts among tasks. However, the results do not sufficiently support the paper’s claims, and the presentation of the methods section requires significant improvement.

**Strengths:**

1) The motivation and framing of LIPO is novel, essentially to find the Pareto optimal solutions that achieve the best quality-length constraint.

2) Extensive experiments on ten reasoning datasets.

**Weaknesses:**

1) The introduction moves from discussing the difficulty of improving small multimodal models (SLMs) to reinforcement learning for reasoning, and then to model merging, without a coherent narrative linking these ideas. The central motivation is not clearly stated — it’s unclear whether the focus is on improving reasoning efficiency, reducing model size, or combining task expertise. The authors should better articulate the overarching problem and explicitly explain how RL and model merging together address it, to create a smoother and more logically connected introduction.

2) The motivation and justification for the LIPO formulation are unclear. The adjusted reward in equation 3 and the weighting in equation 4 are introduced without explaining their design choice. Also, why they are separated instead of being integrated into a single update rule. The choice of Lopt as the maximum of twice the minimum length and the median length seems arbitrary and is not justified. The paper also does not explain how this formulation ensures Pareto-optimal solutions, as no analysis or evidence is provided to show that the learned policy approaches the Pareto front in the quality–length trade-off.

3) The derivation and rationale for the alpha and beta terms in AMM are unclear. The paper defines alpha as a task-importance weight based on the norm of each task vector and beta as a dynamic compatibility factor, but does not explain how these specific formulations were chosen or whether they were derived from any theoretical principle. The same goes for the penalty term R.

4) Overall, the presentation of the methods section needs to be improved.

5) The link between LIPO and AMM is unclear. While LIPO focuses on finding Pareto-optimal solutions (best quality-length trade-offs) and AMM on mixing different experts, the paper does not explain how the two components interact or influence each other. It seems that they operate independently.

6) The paper does not compare model merging with alternative unified training strategies. It is unclear how AMM’s performance and efficiency differ from simply combining all datasets and running GRPO or LIPO jointly, or from sequentially applying GRPO and LIPO across tasks. Without such comparisons, it is difficult to assess whether model merging provides any real advantage over direct multi-task or sequential reinforcement learning approaches.

7) The claim that Tiny-R1V uses less training data than models like InternVL2.5-2B and VITA-1.5-8B is not well supported. Since Tiny-R1V is obtained by merging task-specific experts trained separately on math, structure, and OCR datasets, the total data exposure across all experts should be comparable to training a single unified model. The paper does not clarify whether data overlap or reuse occurs, or how the total training data volume is measured, making the “less data” claim questionable.

8) The response length comparison is not an apples-to-apples evaluation. The paper compares models trained with LIPO to other works using GRPO, but those baselines may differ in architecture, training setup, or data, making the comparison inconclusive. A fair test would require training Tiny-R1V itself with GRPO under identical conditions. Moreover, these results do not validate the claim that LIPO achieves Pareto-optimal behavior as stated in the methods section, since no analysis demonstrates that the model’s responses lie on or approach a Pareto frontier.

9) The AMM ablation results show small gains of 1.1% and 0.3%, but the paper does not indicate whether these differences are statistically significant.

10) Providing standard deviation or confidence intervals would help determine the robustness of these reported gains.

11) The role of the length threshold and reward threshold in LIPO is not clearly explained in methods section (according to me), making it difficult to interpret their impact in "Hyperparameter studies of LIPO".

12) The qualitative result can be moved to the appendix. More importantly, the claim that LIPO attains Pareto-optimal solutions is not empirically demonstrated; stronger experiments are needed to show a clear quality–length Pareto frontier. This space can be used for this analysis.

Overall, the need for model merging after obtaining the individual LIPO-trained models is not well justified. If task vectors can be merged effectively through AMM, a unified model trained on all datasets using LIPO should, in principle, learn the same shared representations without the additional merging step. It is unclear why the authors did not attempt to perform LIPO directly on a combined dataset with phased task batches. As presented, the benefits of combining LIPO and AMM remain unconvincing and appear to add unnecessary complexity without clear empirical or theoretical justification.

Perhaps, I could be missing any details and would be happy to raise my score if the authors address my concerns.

**Questions:**

Please see Weaknesses.

---

### Official Review · Reviewer_TB5h · 2025-11-01

**Soundness:** 3
**Presentation:** 3
**Contribution:** 2
**Rating:** 4
**Confidence:** 3

**Summary:**

The paper proposes Tiny-R1V, a 3B-parameter lightweight multimodal reasoning model trained via a two-stage framework: (1) Length-Informed Relative Policy Optimization (LIPO) to incentivize concise yet accurate reasoning, reducing token usage during RL; (2) Adaptive Model Merging (AMM) to merge task-specific experts (math, structured data, OCR) into a unified model without additional data, using dual adaptive weights and a gradient projection regularization.
LIPO introduces adaptive reward adjustment and dynamic, length-aware advantage estimation, prioritizing shorter responses with similar reward quality. It achieves large reductions in average response length while maintaining or improving accuracy.
AMM adaptively weights task vectors based on inherent importance (α) and dynamic compatibility (β), and regularizes updates to align with task-vector directions, mitigating interference across tasks.
Across 10 benchmarks (math, structured data, OCR, general), Tiny-R1V-3B delivers state-of-the-art performance among lightweight/open models and surpasses multiple merging baselines (e.g., WUDI) and several larger/similar-size MLLMs, with significantly fewer tokens and faster inference.

**Strengths:**

Novel RL formulation (LIPO) that explicitly internalizes response length into reward shaping and advantage estimation, yielding substantial token savings (e.g., 3–4× reduction vs GRPO on MathVision) without sacrificing accuracy.

Clear, principled design: length-triggered reward boosts with smooth decay, weighted advantage using group-optimal length, and compatibility with rule-based rewards (exact match, Levenshtein).

Training-free unification (AMM) with dual adaptivity (α, β) and gradient projection regularization to reduce task conflict; demonstrated improvements over strong merging baselines (e.g., +0.8 avg over WUDI).

Comprehensive evaluation on diverse reasoning tasks (math, tables/charts/documents, OCR) plus general benchmarks (MME, MMStar), with consistent gains and ablations isolating contributions (LIPO hyperparameters, AMM components).

**Weaknesses:**

Limited comparison with mixture training: The current “model merging vs. mixture training” comparison is relatively coarse. Table 2 shows their average scores are very close on most datasets, and mixture training, with more systematic hyperparameter and data-mixing searches (e.g., task sampling probabilities, loss weighting, curriculum/staged mixing strategies), could yield a more robust or higher upper bound in a unified model. This makes it hard to delineate the advantage boundary and applicability conditions of model merging relative to mixture training.


Generalizability to larger models, more tasks, and other architectures: Experiments are validated only at the 3B scale on a single backbone (Qwen2.5-VL-3B), lacking extrapolation to larger parameter counts (7B/13B+), different backbones, and a greater number of task experts (>3 domains).


Trade-off between reasoning length and task difficulty: While LIPO reduces tokens via length-aware advantage reweighting, there is insufficient safeguards regarding potential systematic errors from “premature truncation/under-reasoning,” especially on tasks that may require longer chains, considering the task in this work is relatively not challenging.


Insufficient sensitivity analysis of AMM hyperparameters: The sensitivity of AMM’s hyperparameters across tasks, numbers of experts, and scales has not been systematically presented. The tuning cost and stability risks for practical reproduction and extension to new tasks/models remain unknown.

**Questions:**

See Weakness.

---

### Official Review · Reviewer_tFs8 · 2025-11-03

**Soundness:** 2
**Presentation:** 2
**Contribution:** 1
**Rating:** 2
**Confidence:** 4

**Summary:**

This work attempts to simultaneously address multi-task compatibility and efficient chain-of-thought (CoT) reasoning. The authors propose a method called LIPO that shortens CoT during RL training via Adaptive Reward Adjustment and Dynamic Advantage Estimation. In addition, they train multiple models on different tasks and then merge them into a single multi-task model using a variant of the wudi merging approach.

**Strengths:**

The paper identifies a key issue in contemporary reasoning models, i.e., redundant CoT length and seeks to mitigate it at training time through reward design and advantage estimation while maintaining performance. It also proposes a model-merging strategy to resolve conflicts arising from multi-task consolidation.

**Weaknesses:**

1. The contribution appears incremental. First, LIPO essentially aims to assign higher rewards to shorter responses among similarly high-quality outputs (Adaptive Reward Adjustment) and to weight advantages higher when the CoT length is closer to an ideal length (Dynamic Advantage Estimation). However, because the method uses GRPO, most tasks (except the OCR task that uses Levenshtein distance as the reward) receive binary rewards (0/1). This effectively reduces the training to preferring trajectories with shorter (median-length) CoT that are correct, which resembles common strategies for shortening CoT length. Second, the paper introduces model merging to consolidate models trained on different tasks; although the approach improves upon wudi, multi-task merging, especially in the multimodal LLM (MLLM) setting, has already been explored, e.g., [1], directly targeting multi-modal, multi-task model merging consistent with this paper’s objectives.
2. The design of Adaptive Reward Adjustment is unclear in terms of practical value. If the goal is simply to induce shorter CoT, one can directly constrain response length (e.g., truncating responses beyond a maximum length as in [2]) or incorporate a prior ideal length and smooth the reward to optimize toward that length as in [3]. Both strategies have been shown to yield efficient CoT with strong performance, which raises questions about the added value of the proposed design.
3. The method requires training three separate models and then merging them. Compared with training a single model on the full dataset (or even training only on math data), this yields only ~1% performance improvement.
4. The paper lacks comparisons against existing MLLM reasoning models and validation on larger model scales, such as 7B.

[1] Unifying Multimodal Large Language Model Capabilities and Modalities via Model Merging. arxiv2505.

[2] Vision-r1: Incentivizing reasoning capability in multimodal large language models. arxiv2503.

[3] Dapo: An open-source llm reinforcement learning system at scale. arXiv2503.

**Questions:**

N/A

---

### Note · Authors · 2025-11-12

I have read and agree with the venue's withdrawal policy on behalf of myself and my co-authors.